# Views of the General Population on Newborn Screening for Spinal Muscular Atrophy in Japan

**DOI:** 10.3390/children8080694

**Published:** 2021-08-12

**Authors:** Tomoko Lee, Sachi Tokunaga, Naoko Taniguchi, Tetsuro Fujino, Midori Saito, Hideki Shimomura, Yasuhiro Takeshima

**Affiliations:** Department of Pediatrics, Hyogo College of Medicine, 1-1 Mukogawa-cho, Nishinomiya 663-8501, Hyogo, Japan; sa-tokunaga@hyo-med.ac.jp (S.T.); nao-taniguchi@hyo-med.ac.jp (N.T.); the-jetset@hotmail.co.jp (T.F.); tanmido-sai@hotmail.co.jp (M.S.); shimomura.ped@gmail.com (H.S.); ytake@hyo-med.ac.jp (Y.T.)

**Keywords:** spinal muscular atrophy, newborn screening, general population, public intention

## Abstract

Spinal muscular atrophy (SMA) is a genetic neuromuscular disorder that results in progressive muscle atrophy and weakness. As new therapies for SMA have been developed, newborn screening for SMA can lead to early diagnosis and treatment. The objective of this study was to gather the general population’s view on screening of SMA in newborns in Japan. A questionnaire survey was conducted on two general population groups in Japan. A total of 269 valid responses were obtained. In the general population, about half of the participants had no knowledge about SMA, and more than 90% did not know about new therapies for SMA. Conversely, more than 95% of the general population agreed with screening newborns for SMA because they believed that early diagnosis was important, and treatments were available. This study revealed that the general population in Japan mostly agreed with screening for SMA in newborns even though they did not know much about SMA. Newborn screening for SMA is promising, but it is in very early stages. Therefore, SMA newborn screening should be performed with sufficient preparation and consideration in order to have a positive impact on SMA patients and their families.

## 1. Introduction

Spinal muscular atrophy (SMA) is an autosomal recessive neuromuscular disorder caused by mutations in the *survival motor neuron 1* (*SMN1*) gene, affecting approximately 1 in 6000 to 10,000 live births [1]. Approximately 95–98% of SMA patients have deletions of both *SMN1* gene alleles, while 3–4% carry compound heterozygous point mutations with deletion [2]. *SMN2*, a paralog of *SMN 1*, is known as an important disease-modifying gene, as a greater copy number tends to be associated with a milder clinical phenotype [3]. SMA is characterized by the degeneration of motor neurons in the spinal cord, resulting in progressive muscle atrophy and weakness. Depending on the age of onset and the highest motor milestones achieved, SMA is traditionally classified into clinical subtypes (0–IV) [4,5]. Type 0 is the most severe phenotype and has a prenatal onset. The most frequent phenotype is type I, manifesting in the first six months of life, and results in the child being unable to sit without support. Most type I infants die before the age of 2 years old because of respiratory failure without mechanical ventilation [6]. SMA is the leading genetic cause of death in infants [7].

SMA is considered an incurable disease. However, in addition to improvements in supportive care, treatment for SMA has dramatically improved in recent years, and finally, three SMA-modifying therapies have been approved: nusinersen (Spinraza^®^) (2016 in the USA, 2017 in Europe and Japan), onasemnogene abeparvovec-xioi (Zolgensma^®^) (2019 in the USA, 2020 in Japan, the Middle East, and Europe), and risdiplam (2020 in the USA, 2021 in Europe) [4,8,9,10,11]. Early intervention, particularly pre-symptomatic treatment, showed significantly better motor development [12,13,14]. Treatment should be introduced when the motor neurons are still viable; however, approximately 95% of motor neurons are lost by 6 months of age in type I patients [15]. The importance of early diagnosis is being emphasized, as early diagnosis results in the early initiation of treatment.

Newborn screening has been widely performed worldwide to identify potentially fatal and disabling diseases. Early diagnosis through newborn screening enables early intervention, resulting in a better prognosis. As new therapies for SMA have been developed, newborn screening for SMA is expected to enable pre-symptomatic diagnosis and early treatment [16]. Pilot studies of SMA newborn screening have already begun in some areas [17,18,19,20] as well as in Japan [21].

The introduction of newborn screening for SMA remains controversial. Therefore, the objective of this study was to gather the general population’s view on SMA newborn screening in Japan. 

## 2. Materials and Methods

### 2.1. Survey Design

In order to reveal parent intentions in the general population regarding SMA newborn screening, a questionnaire survey was conducted on two groups (Figure 1). In the first group, the participants were guardians whose children visited the outpatient allergy clinic at Hyogo College of Medicine College Hospital between May 2020 and July 2020. An explanation sheet and a questionnaire regarding SMA newborn screening was distributed at the same time, and responses were obtained anonymously. An explanation sheet briefly described SMA, including clinical symptoms, diagnosis, treatment, and the possibility of newborn screening. In the second group, participants were parents of 5-year-old children who were at Inagawa-cho, Hyogo for a medical checkup. As mentioned above, a brief explanation sheet and a questionnaire were mailed, and responses were mailed back. 

In addition, this survey was conducted with parents of SMA patients who were treated by nusinersen at Hyogo College of Medicine College Hospital between August 2020 and May 2021 (Figure 1).

A questionnaire survey was conducted on two groups of the general population. We included 269 (Group 1:100, Group 2:169) responses, excluding 13 invalid responses. A total of eight responses from the parents of SMA patients were also included. 

### 2.2. Statical Analysis

Statistical analysis was performed using JMP Pro 14 software for windows (SAS Institute Inc., Cary, NC, USA). Fisher’s exact test was used to compare differences in opinion depending on knowledge about SMA. The difference was deemed statistically significant at *p* < 0.05.

### 2.3. Ethics

Participation in this survey was voluntary and only upon the agreement of the individual. This study was approved by the Ethics Committee of the Hyogo College of Medicine, Japan (approval no. 3489). 

## 3. Results

### 3.1. Results in General Population

The total number of valid responses was 269 (Group 1:100, Group 2:169), excluding 13 responses where three or more answers were found to be incomplete (Figure 1). The characteristics of the participants are shown in Table 1. Of the 269 participants, the majority were mothers (94.4%) and were between 30 to 39 years of age (62.5%). All of the participants except for two had no relationship with any SMA patients.

The questions and answers in this survey are shown in Figure 2. Approximately half of the participants had no knowledge relating to SMA before the survey (Figure 2a). Furthermore, most participants did not know about new treatments for SMA (Figure 2b). Nevertheless, more than 95% of participants preferred newborn screening for SMA (Figure 2c). Although participants with more knowledge about SMA tended to favor SMA newborn screening, there was no significant difference (*p* = 0.0566). The answers regarding costs for SMA newborn screening varied; some participants would have approved of testing regardless of price, while others would have done so only if it was free of cost (Figure 2d). When asked about the reason for preferring SMA newborn screening, many respondents considered that available treatment and early diagnosis are important (Figure 2e). A few individuals did not prefer screening. They felt anxious about the test or considered it unnecessary to have a pre-symptomatic test (Figure 2f).

### 3.2. Results in Parents of SMA Patients

All SMA patients who were treated with nusinersen at our hospital during 2020 to 2021 were included. All eight participants agreed with the SMA newborn screening (Figure 3a). They wished for testing regardless of the price (62.5%) or at the cost of ~10,000 yen (37.5%), indicating an aggressive attitude (Figure 3a,b). The main factors for the support of testing are that “early diagnosis is important” and “treatments are available” (Figure 3c). In addition, the participants provided the following responses: “an early treatment can slow the progression of the disease” and “pre-symptomatic diagnosis can remove the baby’s suffering earlier.” 

## 4. Discussion

This research was conducted to assess whether SMA newborn screening would be acceptable to the general population in Japan. The results showed that more than 95% of the general population agreed with newborn screening for SMA, and the key reasons for supporting SMA newborn screening were that early diagnosis is important and that treatments are available. The percentage of patients in favor of SMA newborn screening in the present study was the same as or even higher than in previous studies performed in other countries. In the UK survey, 84% of participants favored screening newborns for SMA [22]. In the USA, 79% of participants agreed even if there was no treatment available, and up to 87% agreed with newborn screening if a treatment for SMA existed [23]. These previous reports highlighted that early diagnosis was considered the most significant benefit of SMA newborn screening in the general population, even if no effective treatments were available, because it would enable parental acceptance of the condition and future pregnancy planning. 

Conversely, early identification of SMA can result in a paradoxical reaction. In the present study, a few participants did not agree with SMA newborn screening because they thought that testing would make parents anxious, and testing should only be performed after symptoms appear. In addition, some participants might be worried that identifying SMA soon after birth will have a negative impact on the parent–child relationship, diminish enjoyment of disease-free time, and cause anxiety about the future. However, a previous study showed that 74% of the general population thought that pre-symptomatic identification of SMA at birth would not prevent families and children from enjoying life while they are symptom-free, and 87% thought that it would not interfere with the early bonding process between parents and children [22]. The fact that most of the participants in the present study were mothers, which could have affected the results. Fathers and mothers may differ in anxiety about SMA screening. When screening is performed, the fact that some parents may feel anxious should be taken into consideration. To relieve parental anxiety and to prevent negative impacts, pre-testing explanation and post-diagnosis care must be enforced as well as proper genetic counseling. 

In the present and previous studies, families of patients with SMA also regarded early identification by newborn screening as a significant benefit [22]; this might be because they had high knowledge of SMA, and they had already experienced the long SMA diagnosis process, including painful examinations. Our results showed that the families of children treated with nusinersen strongly supported SMA newborn screening, which was probably due to the development of new treatments. 

In Germany, a pilot screening project for SMA was performed between 2018 and 2019. In total, 165,525 children were screened, and 22 children with SMA were identified [19]. In that project, based on the recommendations of the American SMA NBS-Multidisciplinary Working Group [24], early treatment with nusinersen was recommended for all children with two or three *SMN2* copies, and strict follow-up monitoring was recommended for children with four or more copies. A total of 10 out of 14 patients with two or three *SMN2* copies were treated with nusinersen at an exceedingly early time point (15–39 days). Although the follow-up duration was short, pre-symptomatically treated patients showed good outcomes, indicating that SMA newborn screening enabled early diagnosis and pre-symptomatic treatment, leading to a good prognosis. However, this project also revealed some problems during SMA newborn screening. First, a few patients did not follow the medical advice concerning treatment or follow-up. Second, it was difficult to predict the clinical phenotype. The number of *SMN2* copies is a major SMA modifier, but it does not always correlate with the clinical phenotype of each patient. In particular, the handling of patients with four or more *SMN2* copies is a difficult problem [25,26], and among patients with four copies, when and for whom treatment should be initiated is currently under debate. 

Another important result of the present study was that most of the general population was unaware of SMA. Although they did not know much about the disease or treatment, they tended to support SMA newborn screening. Ideally, participants would have sufficient prior knowledge of SMA to make an informed decision about whether to accept screening for this condition. For this purpose, we need to increase public awareness of SMA. 

As SMA newborn screening is in a very new ongoing trial, there is little preliminary data. The long-term outcomes of pre-symptomatically treated SMA patients are unknown. The future prognosis of patients with four or more *SMN2* copies has not yet been determined. Hence, it is necessary to accumulate more data to determine appropriate treatment indications and to build a follow-up system. Therefore, when SMA newborn screening is performed, there should ideally be an established flow of treatment, a long-term follow-up system, support for the family, and a subsidy for medical expenses already in place. Based on the results of the present study, SMA newborn screening should be made meaningful for SMA patients and their families. 

## 5. Conclusions

This study showed that the general population in Japan mostly agreed with SMA newborn screening. SMA newborn screening is promising, but it is still in its very early stages. Therefore, SMA newborn screening should be performed with sufficient preparation and consideration to create a positive impact on SMA patients and their families. 

## Figures and Tables

**Figure 1 children-08-00694-f001:**
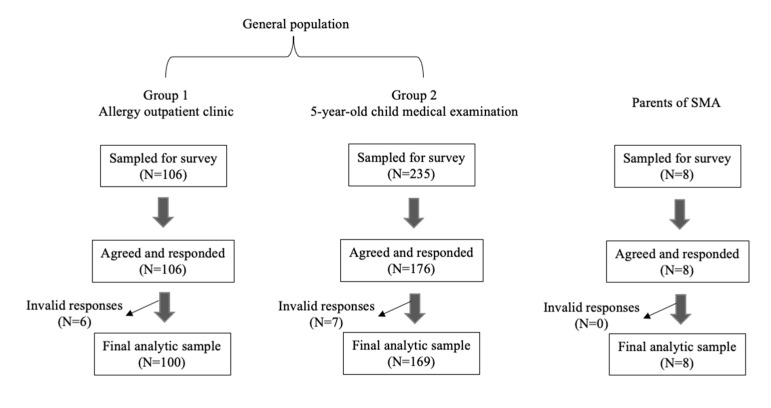
Survey design. SMA: Spinal muscular atrophy.

**Figure 2 children-08-00694-f002:**
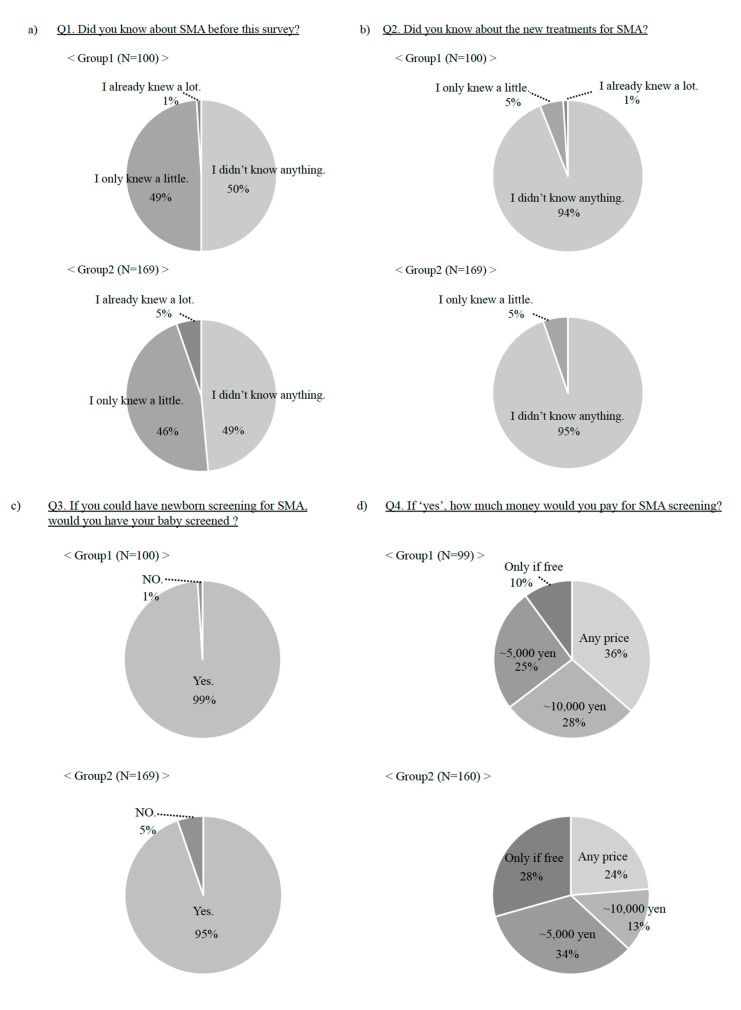
Questions and answers of the general population. Questions (Q1~Q6) and answers of the general population are shown.

**Figure 3 children-08-00694-f003:**
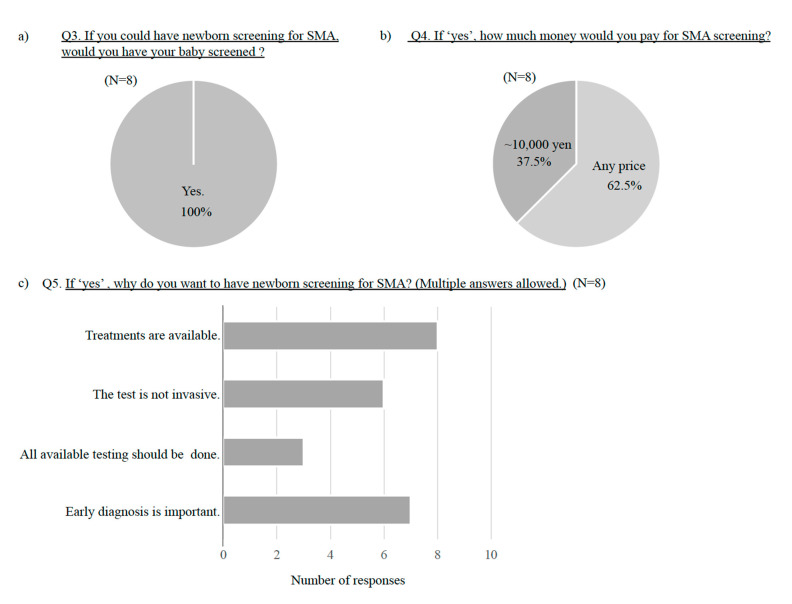
Questions and answers from parents of patients with SMA. Questions (Q3–Q5) and answers of parents of patients with SMA are shown.

**Table 1 children-08-00694-t001:** Respondent characteristics in general population.

Characteristics		All *n* = 269 (%)	Group 1: Outpatient Allergy Clinic *n* = 100 (%)	Group 2: 5-Year-Old Checkup *n* = 169 (%)
Relationship with the child	Father	4.8	7.0	3.6
	Mother	94.4	91.0	96.4
	Grandparent	0.4	1.0	0
	Not specified	0.4	1.0	0
Age (years)	20–29	3.7	6.0	2.4
	30–39	62.5	53.0	68.0
	40–49	32.0	38.0	28.4
	Over 50	1.9	3.0	1.2
Is the respondent close with an SMA patient?	Yes	0.7	2.0	0
	No	99.3	98.0	100

SMA: Spinal muscular atrophy.

## Data Availability

The data presented in this study are available upon request from the corresponding author.

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
