# Peer review of "Views of the General Population on Newborn Screening for Spinal Muscular Atrophy in Japan"

_children, 2021, doi:10.3390/children8080694_

Round 1
Reviewer 1 Report
As a biologist I have impression that the answer of parents about the NBS screen could be influenced by the SMA fact sheet they have received. It would be interesting for me to check whether the parents would be in favor of SMA NBS (and / or will they pay for the testing) before getting the explanation about the disease and whether they change their answer after getting it. It would be also interesting to compare different populations as for their knowledge about SMA and opinion about SMA NBS.
Author Response
Thank you very much for your time reviewing our manuscript and for giving us your valuable comments. Based on your comments, we have revised this manuscript. The amended parts are written in red in the attached manuscript.
As a biologist I have impression that the answer of parents about the NBS screen could be influenced by the SMA fact sheet they have received. It would be interesting for me to check whether the parents would be in favor of SMA NBS (and / or will they pay for the testing) before getting the explanation about the disease and whether they change their answer after getting it.
-Thank you for your valuable suggestion. We agree with your comment that it would be interesting to check the parents’ opinion before the explanation and how their opinion changes after the explanation. We apologize but in the present study we did not collect this data.
It would be also interesting to compare different populations as for their knowledge about SMA and opinion about SMA NBS.
-We appreciate your suggestion. As per your suggestion, we analyzed to compare opinions based on the knowledge about SMA. Statistical analysis was added in Page 3 Line 83-87. Although participants with more knowledge about SMA tended to favor SMA newborn screening, there was no significant difference (p=0.0566). This result was added (Page 3, Line 111-112).
Reviewer 2 Report
The Authors investigated the general population’s view on newborn screening of SMA at the Hyogo Center of Medicine College Hospital between May and July 2020.
The article is well written, and the results are clearly presented. No major revisions are needed.
Minor revisions:
- Line 43: as the Authors mentioned, life span and quality of life have dramatically improved in SMA patients in the recent years. Please consider adding some updated guidelines for the management of SMA patients such as “Diagnosis and management of spinal muscular atrophy: Part 1: Recommendations for diagnosis, rehabilitation, orthopedic and nutritional care. Neuromuscular Disorders 28 (2018) 103–115” and “Diagnosis and management of spinal muscular atrophy: Part 2: Pulmonary and acute care; medications, supplements and immunizations; other organ systems; and ethics. Neuromuscular Disorders 28 (2018) 197-207”.
- Line 134: when discussing the anxiety feeling associated to SMA newborn screening, please consider mentioning the fact that the 94.4% of you sample size is composed by mothers. Since anxiety disorder is higher in women, this result might change in a population with more equal distribution between men and women.
Author Response
Thank you very much for your time reviewing our manuscript and for giving us your valuable comments. Based on your comments, we have revised this manuscript. The amended parts are written in red in the attached manuscript.
Minor revisions:
Line 43: as the Authors mentioned, life span and quality of life have dramatically improved in SMA patients in the recent years. Please consider adding some updated guidelines for the management of SMA patients such as “Diagnosis and management of spinal muscular atrophy:
Part 1: Recommendations for diagnosis, rehabilitation, orthopedic and nutritional care.
Neuromuscular Disorders 28 (2018) 103–115” and “Diagnosis and management of spinal muscular atrophy: Part 2: Pulmonary and acute care; medications, supplements and immunizations; other organ systems; and ethics. Neuromuscular Disorders 28 (2018) 197-207”.
-Thank you for your valuable suggestion. As per your suggestion, we have revised this part as follows (Line 39-44). ‘However, in addition to improvement in supportive care, treatment for SMA has dramatically improved in recent years, and finally, three SMA-modifying therapies have been approved: nusinersen (Spinraza®️) (2016 in the USA, 2017 in Europe and Japan), onasemnogene abeparvovec-xioi (Zolgensma®️) (2019 in the USA, 2020 in Japan, the Middle East, and Europe), and risdiplam (2020 in the USA, 2021 in Europe) [4,8-11]’. These two guidelines were added (reference # 10, 11).
Line 134: when discussing the anxiety feeling associated to SMA newborn screening, please consider mentioning the fact that the 94.4% of you sample size is composed by mothers. Since anxiety disorder is higher in women, this result might change in a population with more equal distribution between men and women.
-We appreciate your comments. As per your comments, the following sentences were added in discussion part (Page 6, Line 161-163). ‘The fact that most of participants in the present study were mothers could affect the results. Fathers and mothers may differ in anxiety about screening.’